# Influence of Tool Geometry and Process Parameters on the Properties of Friction Stir Spot Welded Multiple (AA 5754 H111) Aluminium Sheets

**DOI:** 10.3390/ma14051157

**Published:** 2021-03-01

**Authors:** Danka Labus Zlatanovic, Sebastian Balos, Jean Pierre Bergmann, Stefan Rasche, Milan Pecanac, Saurav Goel

**Affiliations:** 1Department of Production Engineering, Faculty of Technical Science, University of Novi Sad, 21000 Novi Sad, Serbia; danlabus@uns.ac.rs (D.L.Z.); sebab@uns.ac.rs (S.B.); 2Department of Production Technology, Technische Universität Ilmenau, 98693 Ilmenau, Germany; jeanpierre.bergmann@tu-ilmenau.de (J.P.B.); stefan.rasche@ymail.com (S.R.); 3School of Engineering, London South Bank University, London SE1 0AA, UK; goeLs@Lsbu.ac.uk; 4School of Aerospace, Transport and Manufacturing, Cranfield University, Cranfield MK43 0AL, UK; 5Department of Mechanical Engineering, Shiv Nadar University, Gautam Budh Nagar 201314, India

**Keywords:** friction stir spot welding, multiple ultrathin sheets, tool geometry, 5754-H111 aluminium alloy, small punch test, scanning acoustic microscopy

## Abstract

Friction stir spot welding is an emerging spot-welding technology that offers opportunities for joining a wide range of materials with minimum energy consumption. To increase productivity, the present work addresses production challenges and aims to find solutions for the lap-welding of multiple ultrathin sheets with maximum productivity. Two convex tools with different edge radii were used to weld four ultrathin sheets of AA5754-H111 alloy each with 0.3 mm thickness. To understand the influence of tool geometries and process parameters, coefficient of friction (CoF), microstructure and mechanical properties obtained with the Vickers microhardness test and the small punch test were analysed. A scanning acoustic microscope was used to assess weld quality. It was found that the increase of tool radius from 15 to 22.5 mm reduced the dwell time by a factor of three. Samples welded with a specific tool were seen to have no delamination and improved mechanical properties due to longer stirring time. The rotational speed was found to be the most influential parameter in governing the weld shape, CoF, microstructure, microhardness and weld efficiency. Low rotational speeds caused a 14.4% and 12.8% improvement in joint efficiency compared to high rotational speeds for both tools used in this investigation.

## 1. Introduction

An emerging trend in the automotive and aerospace industry is to use lightweight alloys based on aluminium, magnesium and titanium or advanced high strength steels to reduce the vehicle weight. Thereby, weight reduction will potentially lead to a reduction of fuel consumption, greenhouse gas emissions and reduced energy consumption during production [1,2,3]. Aluminium is recognised as a favourable conductor material due to its low weight, relatively low and stable cost, high corrosion resistance and good electrical conductivity. Furthermore, friction stir welding (FSW) and friction stir spot welding (FSSW)have become revolutionary welding techniques over the past two decades because of energy efficiency, environmental friendliness and high-quality joints [4]. According to Fastmarkets [5], the price of aluminium is almost four times lower than that of copper. Also, aluminium has a 3.3 times lower density than copper and slightly higher specific strength [6]. Also, its thermal conductivity is lower compared to copper (386 vs. 237 W/m K) but still much higher than that of carbon steel (54 W/m K) or 316 L and 304 stainless steel (15 and 16.3 W/m K) [7]. Consequently, great research attention is given to substituting copper with aluminium for battery components, stand-thermal connectors, terminals and wire conductors [8,9].

Due to good weldability, corrosion resistance and mechanical properties, aluminium AA 5754-H111 alloy has stood out in various industries such as automotive, aerospace, chemical and nuclear [10,11]. However, the joining efficiency of aluminium alloys is still facing considerable challenges. To move towards efficient manufacturing and to reduce electrical energy consumption and costs, welding technology has to be carefully selected and optimised [12].

Resistance spot welding (RSW) continues to remain a predominant body-in-white process in the automotive industry. However, high electrical energy consumption, as well as the tendency to degrade the electrodes, make this process too expensive for joining aluminium and its alloys [2,12,13]. This has motivated researchers, especially in the automotive industry, to seek novel or improved low-energy consumption joining processes. Friction stir spot welding (FSSW) is a relatively novel technique patented by Mazda Motor Corporation. It is a solid-state lap joining process with low energy consumption that can produce defect-free spot welds with better mechanical properties compared to RSW [12,14]. Moreover, FSSW does not use consumables, shielding gases, electrodes, or cooling liquids, which makes this joining process more environmentally friendly compared to the traditional RSW [15]. Finally, the additional drawback of the RSW joining process is that the electrodes are based on copper and contain various alloying elements. Typical alloying elements are tungsten, cobalt, beryllium and silicon metal, which are critical raw materials (CRMs), while chromium, zirconium and molybdenum are near critical elements [16]. The degradation and consumption of these electrodes, especially for aluminium welding due to the high contact resistance induced by the presence of resistant aluminium oxide layer on the surface of Al alloys is very high [17]. Therefore, FSSW can lead to a reduction in the consumption of critical raw materials during the spot-welding techniques.

During FSSW, the non-consumable rotational tool penetrates the topmost sheet of the lap joint, while the backing plate supports the bottom most sheet. The weld efficiency depends on process parameters such as the axial load, rotational speed, penetration depth, dwell time and tool geometry [18,19,20]. During the FSSW process, the material beneath the tool is stirred and plastically deformed, which causes the formation of a metallurgical bond. However, during conventional FSSW after retrieval of the tool, a key-hole is left beneath the tool which compromises the mechanical properties of the weld [21]. Several modified FSSW techniques are developed to overcome the formation of key-hole such as the Refill FSSW [22], Swing FSSW [23], Flat FSSW [24], Friction bit joining [25], walking FSSW [26], double-sided FSSW [27] and pinless FSSW [28,29]. Welding processes such as Refill FSSW, Swing FSSW, Flat friction stir scribe (FSS), Friction bit joining (FBJ), Walking FSSW and Double-side FSSW are time consumable and expensive processes employing complex equipment. A pinless FSSW can provide high-quality keyhole-free joints with minimal dwell time at lower processing cost.

For the massive production of spot welds, a key factor is a dwell (welding) time. The key feature of FSSW that needs to be addressed is its relatively long dwell time varying from 2 to 5 s. Bakavos et al. [30] studied the influence of tool geometry and dwell time on the mechanical properties of AA 6111-T4 aluminium alloy. It was reported that a novel design of pinless tools can result in high shear strength welds (around 3.4 kN shear load) in 1 mm thick aluminium sheets with short dwell times (less than 1 s). However, reduced dwell time also decreases the strength. Tran et al. [31] found that weld strength increases with increasing the dwell time during FSSW of AA 5754 and AA 7075 aluminium alloys. Shahani et al. [19] pointed out that optimal weld mechanical properties were obtained with a dwell time of 2 s for FSSW of AA 6061-T6. The increase of dwell time beyond 2 s has an irrelevant effect on the mechanical properties.

Thus, the overarching aim of this work was to improve the current understanding of the effect of tool geometry and welding parameters on dwell time, microstructure and mechanical properties of friction stir spot welds. Two pinless convex tools with different tool radii were used for FSSW of four AA 5754 aluminium sheets with 0.3 mm thickness. This novel approach in tool concept and geometry potentially enables to join a higher number of thinner sheets, compared to what’s reported in literature. This is essential for the successful embedding of the FSSW process in highly efficient industrial applications for automotive batteries.

In this paper, the variation of welding parameters (rotational speed and axial load) and tool geometries were analysed to obtain high-strength welds with minimum dwell time. Detailed microstructure analysis with an optical microscope and scanning acoustic microscope of weld configuration and weld defects, was carried out. To understand the influence of process parameters, tool geometry and dwell time on mechanical properties, microhardness tests and small punch tests (SPT) were performed. Also, a light microscope and scanning electron microscope were used to analyse SPT fractured specimens.

## 2. Materials and Methods

The base material for FSSW was cut from 0.3 mm thick rolled commercial AA 5754-H111 sheets. In Table 1, the chemical composition determined by optical emission spectroscopy (ARL 3580, Applied Research Laboratories, Waltham, MA, USA) is given. The sheets were cut to 44 mm × 50 mm dimensions. The mechanical properties of the as-received material such as its tensile strength, yield strength and elongation are given in Table 2.

FSSW experiments were conducted by using a force controlled EJOWELD C50R FSSW device (EJOT GmbH & Co. KG Geschäftsbereich EJOWELD, Tambach-Dietharz, Germany) which can reach a maximum rotational speed of 9000 rpm, a maximum axial load of 8 kN. A maximum dwell time of 5 s was used. A detailed explanation of the experimental setup was presented in [9]. Four thin sheets were lap joined by the application of tools having different geometry, shown in Figure 1. The two tool designs (T1 and T2) with different radii used in this research are made from H13 (X40CrMoV51) hot-work tool steel. The radius of the convex tool has a significant influence on dwell time. The axial load was maintained at 2 kN for batch I and 4 kN for batch II. The axial displacement (h) (penetration depth) made by the rotating tool was maintained at 0.25 mm. The tool geometry, axial load (F) and rotational speed (n) and dwell time (td) with standard deviation are shown in Table 3. Welding with parameters below or above presented values did not provide joint through all four sheets. The dwell time is a function of axial load, rotational speed, penetration depth and material properties as the machine is force-controlled [9]. Each experiment was repeated three times and an average value with the standard deviation was reported.

The non-destructive assessment of weld interiors to examine various defects in the weld zone was tested using a Scanning Acoustic Microscope (SAM) (PVA TePla made, Westhausen, Germany), equipped with H2 PreAmplifier. The FSSW joints were immersed in a water tank (deionised water). To obtain images from acoustic waves, a transducer PT30-6-12.7 with 20 dB gain, 30 MHz frequency, a diameter of 6 mm and a focus distance of 12.7 mm was used. The samples were scanned with A-scan, C-scan and B-scan modes. The tested wavelength was set to be 20,000 ns for all samples. For C-scan, the gate length was held at 50 ns.

The metallographic analysis was conducted on welded joints and fractured specimens after a small punch test that was sectioned across the central line and then cold-mounted. Thereafter, fine grinding and polishing were done with sandpapers (360, 600, 1200 and 2500 grit sizes). The samples were then treated with diamond suspensions (6 and 3 µm) and final polishing involved treating the sample with colloidal silica with an average particle size of 0.05 µm (OP-S) for 3 min. After polishing, electrolytic etching with Barker’s etching agent (5 mL HBF_4_ + 200 mL H_2_O) was done on Struers LectroPol-5 device (40 V, 2 min). The cross-section morphology of the joints, grain size and the fractured SPT were analysed by Zeiss, AxioScope A1 light microscope with AxioCam ICc3 for the etched samples and Zeiss Axio Vert.A1 MAT with AxioCam 105 for the polished samples. The etched samples were analysed with crossed polarized light under sensitive tint. For quantitative analysis of the grain size measurement, ImageJ software was used [32]. Fractured SPT specimens were examined by a JEOL JSM-6460LV scanning electron microscope (SEM) operating at 25 kV. Struers DuraScan 70 instrument with a load of 0.1 kg was used to obtain the Vickers microhardness. The microhardness test was performed according to standard [33] in three lines 0.3, 0.6 and 0.9 mm above the bottom surfaces with a spacing between indentations of 0.3 mm.

Due to the disk-shaped miniature size corresponding to the size of the stir zone, the SPT is well suited to investigate the local mechanical properties of non-uniform materials such as spot welds. The load-displacement curve of the SPT can be used to assess the load-carrying capacity of small specimens prior to failure.

The SPT experiments were performed on a Hegewald & Peschke Inspect Retrofit universal testing machine with a maximum load of 20 kN [34,35]. Rasche et al. [34] developed this customised SPT equipment setup, called the small punch bending test, which was placed at a universal testing machine. The SPT setup geometry was modified compared to the geometry recommended by the European “Code of Practice” CWA 15627:2006 [36], as the specimen is neither clamped nor the vertical movement of the disc edge is prevented. A punching speed of 2 mm/min was selected according to previous studies. The specimens for the SPT were cut from the middle in form of Ø8-0.005 mm disks. To consider all three weld interfaces between the four sheets at once, a specimen thickness of 0.8 mm was chosen both for welded and base material. The specimens were fine grounded up to 2500 grit abrasive paper to reduce the sheet thickness from around 0.95 mm to 0.8 ± 0.01 mm as shown in Figure 2. SPT specimens of the base material were cut and prepared from AA 5754-H111 sheet, 1.2 mm thick. All tests were repeated three times.

## 3. Results and Discussions

### 3.1. Coefficient of Friction (CoF)

The variation in the coefficient of friction (CoF) was calculated for the batch I (Figure 3a) and batch II welds (Figure 3b) using Equation (1) [37]. In recent literature, due to the complexity of laboratory measuring procedures for determining CoF, it is usually assumed that CoF during FSSW is a constant value [38,39]. However, in real conditions, due to variation of temperature and material properties during the welding process, CoF varies significantly. In this paper, the CoF was calculated by using torque and axial load data obtained from the welding machine. Although the axial load was maintained to be fixed at the beginning of the experiments, the real-time measurements showed variations and it was the real-time measurement of the axial load which was used in estimating the CoF. Also, the contact radius of the tool (calculated according to Equation (2) [40]) used in estimating the CoF changed gradually from 0 to 2.73 mm for the batch I tool and from 0 to 3.34 mm for the batch II tool. Therefore, in the CoF equation, the time-depended torque, axial load and contact radius were used, and the results here are presented as time-depended diagrams.
(1)$$\mu \left(t\right)=\frac{\tau }{\sigma_{n}}=\frac{2\cdot T\left(t\right)}{F_{N}\left(t\right)\cdot r\left(t\right)}.$$

*τ*—shear stress (MPa)

*σ_n_*—normal stress (MPa)

*T*(*t*)—torque (N·mm)

*F_N_*(*t*)—axial load (N)

*r*(*t*)—tool contact radius (mm) calculated from equation
(2)$$r\left(t\right)=\sqrt{h\left(t\right)\cdot \left(2R-h\left(t\right)\right)},$$
where *R* is the fixed tool radius (mm), *h*(*t*) is the axial displacement of the tool (penetration depth) (mm).

The values of CoF vary from 0.75 to 2.5 during the welding process in batch I and between 0.6 and 2 in batch II. From Figure 3a,b a big difference in the dwell time can be observed (dwell times in batch I are approximately three times longer than those of batch II). A noticeably shorter dwell time (less than 1.7 s) in batch II classifies tool T2 as a more productive tool than the T1 tool. Farhat et al. [41] and Kumar et al. [42] reported, that during the initial interaction between the tool and material surface, friction is driven by the grains from unprocessed material. As the interaction progresses, the material in the subsurface and on the surface deforms and the hardness increases due to strain hardening of the material competing against thermal softening. Thus, the two opposite driving forces operate against each other to dictate the conditions and microstructure of the material at the end of the process. In samples where the CoF increases at the end of the process (at 1500 rpm in batch I and 1000 rpm in batch II), the strain hardening overcomes thermal softening and frictional properties are governed by properties of the strain hardened metal. In samples where CoF increases at the beginning and then decreases near the end of the process, thermal softening overcame strain hardening after reaching the peak value of the CoF. There are also two specimens (2000 rpm in batch I and 1500 rpm in batch II) where CoF fluctuates without significant increase or decrease, making the strain hardening and thermal softening roughly equally influential in the evolution of kinetic CoF.

### 3.2. Weld Joint Analysis

Figure 4 shows the macrographs of the top view (a), a cross-section of polished (b), and a cross-section of etched (c) sample welded at 1500 rpm welded with T2 tool. In the presented macrograph, two different areas can be observed: edge-chipping area (ECA) and stable contact area (SCA). ECA is the rough ring-shaped area with visible marks of deformation and fracture, while SCA is the smooth surface area in the center of the contact with the tool which is in stable contact with the tool. In ECA, the upper sheet is only partly in contact with the tool due to local deformation of the upper sheet caused by the action of the radial force(Fr) caused by rigid clamping from one side and shear load (Fshear) of the tool causing the material to flow radially from the other side. Basically, during FSSW of the thin sheets, upper sheets on the outer part tend to bend upwards, but they are constrained by the clamping system. As a result, the thermal stresses developradially in the sheets. Due to a continuous downward movement of the tool, the radial stress grows further, which causes the upper sheet to deform upwards and to meet the tool under the load conditions (F_EC_) as shown in Figure 5. In those conditions, the portion of material at the upper surface near the tool is thus smeared away leaving some fractured sites [9,43]. The rotational speed has a significant influence on the size and ratio of SCA and ECA.

Figure 6a shows results obtained by measuring the radii of SCA (stable contact area) and ECA (edge-chipping area) from top surface macrographs and by calculating the ratio between them. In both batches, the same trend was observed. With the increase of the rotational speed the ratio increases as well (r/R). Therefore, the samples welded at higher rotational speeds have a larger share of SCA than ECA. Theoretically, weld thickness (0.95 mm) should be equal within the batch as penetration depth was set to be the same for all samples. However, after measuring the axial thickness of the weld in the centre at polished macrographs (Figure 6b), it was observed that *H* was not constant, and it depends on the rotational speed. Therefore, the amount of extruded material from the centre of the weld zone towards the periphery (due to the centrifugal force of the tool) differs as well. In both batches, an increasing rotational speed was found to decrease *H*. In batch II, those values were closer to the theoretical value than those of batch I. In batch II, the samples welded at rotational speeds from 1000 to 3000 rpm due to a low temperature and high viscosity of the weld exhibited elastic recovery, with the possible influence of the thermal expansion of the tool. Therefore, the measured thickness of the weld was higher compared to the predefined thickness. However, in samples welded at 3500 to 4500 rpm from the batch II and all samples from batch I, the stress-induced material flow was observed to cause low viscosity of the stir zone. When the viscosity of the material drops, centrifugal force tends to extrude the material towards the tool edge.

Dimensional quantification of the stir zone as a function of the rotational speed of the tool is shown in Figure 6. To understand the influence of rotational speed and tool geometry at weld shape, the ratio of the bottom and top diameter of the stir zone (d/D) was calculated and presented in Figure 6c. In both batches, the ratio d/D significantly decreases with the increase of the rotational speed. This sharp geometry transformation can also be described with the taper angle of the stir zone shown in Figure 6d. In both batches, an increase from approximately 50° to 80° was observed. This action can be correlated to the material resistance, that is, the viscosity effect. Thus, the samples welded at low rotational speeds where the viscosity of the stir zone is high, stress is governed towards the bottom of the stir zone and thus produces a cylindrical shaped stir zone, whereas in samples welded at high rotational speeds, local thermal softening of the material causes a high drop in viscosity which extrudes the material up towards the periphery of the tool and causes the shape of the stir zone to be conical.

### 3.3. Scaning Acoustic Microscope (SAM) Analysis

Defects, such as delamination between the sheets, were observed only in batch II with respect to the rotational speed. Therefore, the results presented are focused only on studying the weld quality of samples from batch II concerning the rotational speed which had the highest influence on weld quality.

The results of SAM testing for samples welded at 1000, 2500, and 4500 rpm from batch II are presented in Figure 7, Figure 8 and Figure 9. In Figure 7a, Figure 8a and Figure 9a surface topography maps of the top surface obtained with SAM are presented. A high level of plastic deformation on the upper surface from the centre of the weld to the periphery can be seen. In Figure 7b,d, Figure 8b,d and Figure 9b,d, C-scans of the samples in different gate positions are presented. The material flow pattern of the section near to top surface can be seen in Figure 7b, Figure 8b and Figure 9b. In Figure 7e, Figure 8e and Figure 9e, the entire A-scans are shown supported with higher magnification details in Figure 7f, Figure 8f and Figure 9f. The most indicative C-scans are those in Figure 7d, Figure 8d and Figure 9d because they show the presence of delamination circles inside the weld zone at the intersection of the two last sheets (blue arrow). The white ring around the stir zone represents the unbonded region at the intersection of the last two sheets. The next ring (black with white and grey marks) is the area where edge-chipping on the surface of the sample is present. Due to the limitations of SAM, the area below this rough zone cannot be seen. In all three samples, a different surface topography, the volume of the defects and effective welded area between the two last sheets are present. In the first two welding interfaces, no delamination was found in any of the tested samples which can be observed in cross-sectional B-scans (Figure 7c, Figure 8c and Figure 9c). The only delamination at the interface between the last two sheets can be seen (blue arrow).

Observing the third interface, it can be seen that different rotational speeds produce different sizes of bonded areas and volumes of delamination circles. The size of the bonded area in the third interface decreases while the volume and size of delamination circles increase with the increase of rotational speed. This observation correlates well with the d/D ratio in Figure 6. In samples welded at higher rotational speeds, thermal softening overcomes strain hardening and causes fast local material softening below the tool where the friction-induced temperature is expected to be the highest. The material flows upwards toward the periphery of the contact area between the tool and workpiece. Therefore, due to the low viscosity of locally softened material and the high centrifugal force of the tool, material flow radially and there is not enough stress (pressure) transmitted to the lower part of the stir zone. If the stress is not transmitted to the lower part of the weld, the interface will not stretch enough, and the natural oxide layer on top of the aluminium sheets will not be ready for a homogenous diffusion process to occur. This can impede the diffusion process which can trigger delamination circles at the randomly dispersed interface. In welds obtained at lower rotational speeds, strain hardening dominates over thermal softening and due to a lower temperature expected below the tool, stress is better transferred to the bottom interface. Pressure stretches all interfaces almost equally, providing similar diffusion conditions in all three interfaces.

### 3.4. Microstructural Evaluation

The microstructure evaluations of the weld at different rotational speeds from batch II, comparing all metallographically different zones, are presented in Figure 10. The same trend was observed in the batch I as well. The four different zones can be observed—stir zone (SZ), thermo-mechanically affected zone (TMAZ), heat affected zone (HAZ) and base material (BM).

The microstructure evaluation of the welded samples was done in the middle of the stir zone. The results of grain size measurements for both batches are presented in Figure 11 a,b. A clear influence of rotational speed on grain size and distribution can be seen. All weld samples exhibited significant grain size reduction compared to the base material. In both batches, the grain size significantly decreases at low rotational speeds. However, by comparing two batches, it was found that the grains were coarser in SZs of welds from batch II in the samples welded at 1500 and 3500 rpm. However, the samples welded at 2500 rpm showed the opposite trend. In both batches, the increase of the rotational speed led to an increase in the grain size. The grain size varies from 5 µm in sample welded at 1000 rpm (batch II) up to 24 µm in sample welded at 4500 rpm, which is significantly lower compared to those of the base material (69 µm). The base material was a AA 5754-H111 hot rolled ultrathin sheet in which a subgrain lattice dislocation generated during rolling can remain homogeneously dispersed inside of the grains or they can attain a low energy dislocation structure (LEDS) [44]. During the FSSW process, those grains and subgrains tend to be replaced with fine equiaxed recrystallised grains in the stir zone.

Jata et al. [45] proposed continuous dynamic recrystallization (CDRX) as a driving mechanism for grain refinement during FSSW. CDRX relies on subgrain formation during high strain rate and frictional heating. In various research papers [45,46,47,48], it was suggested that the magnitude of misorientation during FSSW/FSW increases significantly compared to the base material. Therefore, the new grains found in the stir zone are highly misoriented subgrains. The microstructure evaluation of samples welded at different rotational speeds is governed by two processes: (i) grain size reduction triggered by subgrain rotation induced by dislocation glide and (ii) subgrain coarsening which causes numerous subgrains to decrease while the average size of each subgrain increases. It is driven by the reduction of local misorientation associated with intercrystalline recovery. The first case is expected in samples where strain hardening dominates over thermal softening (lower rotational speeds). The second case occurs in welds where thermal softening overcomes strain hardening, causing an increase in grain size compared to those samples welded at lower rotational speeds. Therefore, in the stir zone, dynamic recovery takes place where subgrain growth is associated with the absorption of dislocations into the boundaries [49].

Transition zones are those zones which are located between the stir zone and base material, namely the thermo-mechanically affected zone (TMAZ) and heat affected zone (HAZ) (Figure 10). The size and the shape of the grains within them vary in samples welded at different rotational speeds.

The TMAZ is characterized by a highly deformed structure but it is considered that recrystallization does not occur in this zone due to low strain rate [51]. In welds obtained at low rpm, it is a narrow region with columnar grains orientated in the stir direction. However, in welds obtained at higher rotational speeds, the shape of the grains in TMAZ is more equiaxed. Also, rotational speed affects grain size in TMAZ, which varies from 21 µm (at 1500 rpm) to 36 µm (for 3500 rpm). The size (width) of the TMAZ zone is the most affected parameter. Welds obtained at low rotational speeds showed a narrow TMAZ with a sharp and clear transition between the zones, unlike welds obtained at higher rotational speeds, which have a gradual transition between zones (SZ and HAZ). Other parameters such as axial load, dwell time and tool geometry does not affect the shape of the TMAZ, they only affect grain size.

The HAZ is the zone between TMAZ and BM. According to Mishra et al. [51], HAZ exhibits only the thermal cycle without plastic deformation. The grain shape is similar to those of the base material (BM) but with different sizes. Shen et al. [52] studied welds obtained in AA 6061 sheets with FSW where the grain size in the HAZ was higher compared to BM (22 and 17 µm respectively). However, Sahu et al. [53] reported smaller grains in the HAZ compared to BM. Grain sizes ranging from 53 to 62 µm for different welding conditions in the stir zone were reported whereas grains in BM were of the order of 68 µm. In the present work, the HAZ exhibited a grain size that is larger than in the TMAZ and smaller compared to those of BM (Figure 11). In both batches, the grain size in HAZ increased with increasing rotational speed up to 2500 rpm, while, beyond that, it decreases.

### 3.5. Mechanical Properties

#### 3.5.1. Vickers Microhardness

The results of microhardness tests for both batches are presented in Figure 12a–f. In both batches for all rpm, an increase in hardness in the weld centre can be observed in Figure 12a,d. The most pronounced increase in hardness was detected in samples welded at 1500 rpm (Batch I) and 1000 rpm (Batch II). However, by comparing two batches it is clear that higher hardness values are obtained in the batch I for all welds along all three lines measured. This can be correlated to dwell time (Table 3). In the batch I, dwell time for all welded samples was approximately three times lower compared to those of batch II. The highest hardness was observed in the sample welded with the longest dwell time (Batch I, 1500 rpm, 4.65 s). With the decrease of dwell time, the hardness decreases as well. This can be the consequence of longer material stirring, which caused the accumulation of dislocation to be higher. Also, observing from the upper surface to the lower, a reduction in hardness can be seen. In both batches, the hardness values are lowest at the bottom line, Figure 12c,f and highest in top-line Figure 12a,d. According to all the presented results, the welds at 1500 rpm (Batch I) and 1000 rpm (Batch II) showed higher values of hardness compared to the base material, indicating that strain hardening dominates over thermal softening. On the other side, welds obtained at higher rotational speeds exhibited strong domination of thermal softening governed by the dynamic recovery. The thermal softening is responsible for hardness drop in welds below the values of hardness in the base material. Dynamic recrystallization and dynamic recovery caused strain-hardened base material (AA 5754-H111) to gain a new recovered structure with a lower extent of dislocations.

#### 3.5.2. Small Punch Test

A small punch test was performed as an alternative to the shear tensile test, which is usually used for testing FSSW samples. However, during the shear tensile test, the fracture usually occurs at the outside edge of the stir zone [54], or if the bond is too weak, it follows the weld interface line [55]. To understand the mechanical properties of the weld it is necessary to determine macroscopic strength locally and hence a nanoindentation test would not be a proper test either.

In this paper, SPT was proposed as an alternative test, because spot welds obtained by FSSW closely correspond to specimens used in SPT. Furthermore, SPT is well suited to compare the strength of the welds from both batches obtained at different rotational speeds with those of base material. This will provide a profound analysis of the influence of welding parameters and tool geometry on weld quality.

The load-displacement curves of welds obtained from SPT at different rotational speeds from both batches compared with those of base material are presented in Figure 13a,b. Although three samples were tested in all cases, in Figure 13a,b only load-displacement curves with minimum (energy safest) values are shown. Table 4 presents the rupture loads (maximum loads) obtained by SPT. In both batches, it can be seen that rupture load is in correlation with the microhardness results. Also, welds obtained in both batches exhibiting higher microhardness profiles compared with those of base material, have withstood higher rupture loads (specimens welded at 1500 rpm in the batch I and 1000 rpm in batch II). This confirms the hypothesis, that welds obtained at low rotational speeds are strain hardened. All other samples welded at higher rotational speeds ruptured sooner than the base material as a consequence of the thermal softening.

For a better understanding of the SPT results, weld quality can be described by joint efficiency using the equation below [56]:(3)$$E=\frac{R_{m\;\left(weld\right)}}{R_{m\;\left(BM\right)}}\cdot 100$$
where *E* represents joint efficiency (%); *R_m_*
_(*weld*)_—weld strength (MPa) and *R_m_*
_(*BM*)_—base material strength (MPa).

If the thickness of SPT specimens cut from welds and base material are the same, joint efficiency can be calculated by using maximum load (rupture load):(4)$$E=\frac{F_{m\;\left(weld\right)}}{F_{m\;\left(BM\right)}}\cdot 100$$
where *F_m_*
_(*weld*)_ is the rupture load of the welds (N) and *F_m_*
_(*BM*)_ is the rupture load of the base material (N).

This equation analogous to Equation (2) is based on the empirical observation that the strength of the material Rm correlates linearly with the maximum (rupture) load Fm during SPT [34,35]. Even if the correlation factor is not known, as in this case, the relative change in Rm and hence the joint efficiency can be predicted by means of the SPT results.

Weld efficiency (E) with standard deviations is presented in Table 4. The specimens welded at 1500 rpm in the batch I and 1000 rpm in batch II showed higher strength compared to the base material (weld efficiency was more than 100%), while specimens obtained at higher rotational speeds showed lower strength (weld efficiency was less than 100%).

#### 3.5.3. Structure and Dimensional Evaluation of the Small Punch Test Samples

In Figure 14, the cross-section of base material specimens after SPT and two specimens from both batches welded at lower rotational speeds (1500 rpm from batch I and 1000 rpm from batch II) are shown. Those are the welds where strain hardening was the dominating mechanism. In all three specimens, radial crack typical for SPT specimens was observed. In Figure 15a,b top micrographs of specimens welded at 1500 and 3500 rpm from batch I showing the radial crack orientated in the circumferential direction are shown. The batch I specimen has narrowly opened a circumferentially-oriented crack, which is a typical crack obtained after SPT [57] without any sign where the weld interface is and without any trace of delamination. However, in a specimen from batch II, delamination can be observed. During FSSW of aluminium alloys, an oxide layer can get trapped in the weld interface. To provide unobstructed diffusion between the sheets this layer has to be stretched, fractured and mixed during the stirring as much as possible. As the dwell time is approximately three times longer in the batch I compared to the batch II specimen, a better stirring of the oxide layer has occurred. The delamination that was observed in the SPT specimen occurred along the weld interface leading to the belief that the oxide layer has caused poor bonding and delamination to occur.

In Table 5, the specimen thinning measured from the SPT macrographs in the middle of the stir zone from each specimen in both batches is presented. The results of specimen thinning suggest that the ductility of specimens welded at different rotational speeds differs. The lowest values of specimen thinning can be found in specimens welded at the lowest rotational speeds (15 ± 1.25 at specimens from the batch I welded at 1500 rpm and 21.3 ± 1.25 at specimens from batch II welded at 1000 rpm). This fact also verifies the theory that the strain hardening mechanism is dominant at low rotational speed, leading to an increase in hardness and strength and reduced ductility. This can be also proved with scanning electron microscopy (SEM) analysis of specimens welded at 1500 and 3500 rpm from the batch I in Figure 15c,d. In SEM micrographs shown in Figure 15c,d rupture of specimens at higher magnification can be seen. It can be observed that specimen welded at higher rotational speed (3500 rpm) has more ductile surface with several pits typical for ductile fracture, while in specimen welded at lower rotational speed, the number of pits is lower and the surface of the rupture has a less ductile appearance.

## 4. Conclusions

FSSW with the pinless tool is an emerging spot-welding technology that can be used for joining a wide range of materials at a low processing cost. To improve the productivity of FSSW, the dwell time needs to be reduced by using optimal weld parameters to obtain the desired weld properties. This effort is shown to be linked to the optimal consumption of critical raw materials which are used as consumables during the process. This work aimed to explain new insights into the FSSW of ultrathin multiple sheets to achieve minimal dwell time.

The conclusions drawn from this work can be summarised as follows:
The two types of pinless tools used in this investigation provided weld joints of multiple thin sheets (0.3 mm) of AA 5754-H111 aluminium alloy at various rotational speeds.The kinetic coefficient of friction (CoF) was found to be dependent on the rotational speed used. For example, at low rotational speeds, as a strain hardening effect occurs, the CoF increased up to the end of the process. However, in samples welded at higher rotational speeds thermal softening caused a decrease of CoF over the dwell time.Macrographs and SAM analysis proved no defect in the top two weld interfaces. However, delamination circles were found at the bottom most interface. The volume of delamination circles increases with increasing rotation speed. The welds obtained from one particular tool produced a wider stir zone compared to those obtained with the other tool. In both batches, the shape of the stir zone varies from almost cylindrical at low rotational speed to conical at higher rotational speeds.All samples experienced different grain sizes and shapes of SZ, TMAZ, HAZ and BM. The grain size was highest in based materials (BM) followed by HAZ, TMAZ and finally stir zone: Depending on the rotational speed, different widths of the TMAZ and grain shape inside of it were seen. Grain size decreases with decreasing rotational speed.Microhardness and weld efficiency showed, that in samples welded at low rotational speed, improved mechanical properties were obtained compared to the base material due to a dominant strain hardening mechanism while in samples welded at higher rotational speed thermal softening caused a reduction in the mechanical properties, particularly fracture strength. Low rotational speeds caused a 14.4% and 12.8% increase in joint efficiency compared to high rotational speeds for the two types of tools used in this study.


RSW is still being the most productive technology from productivity aspects. However, a number of disadvantages, among which the potential logistical difficulties that may arise from the application of a number of CRMs and near CRMs, can influence the introduction of FSSW as an emerging spot-welding technology.

## Figures and Tables

**Figure 1 materials-14-01157-f001:**
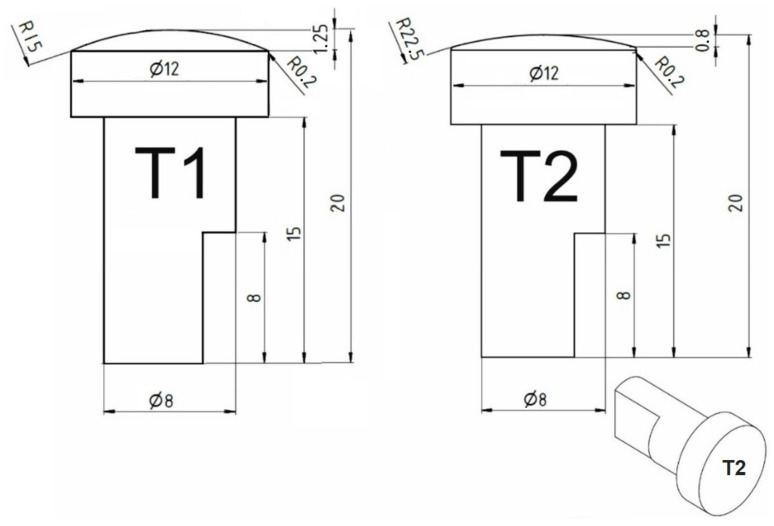
Details of friction stir spot welding (FSSW) tool geometries in (mm).

**Figure 2 materials-14-01157-f002:**
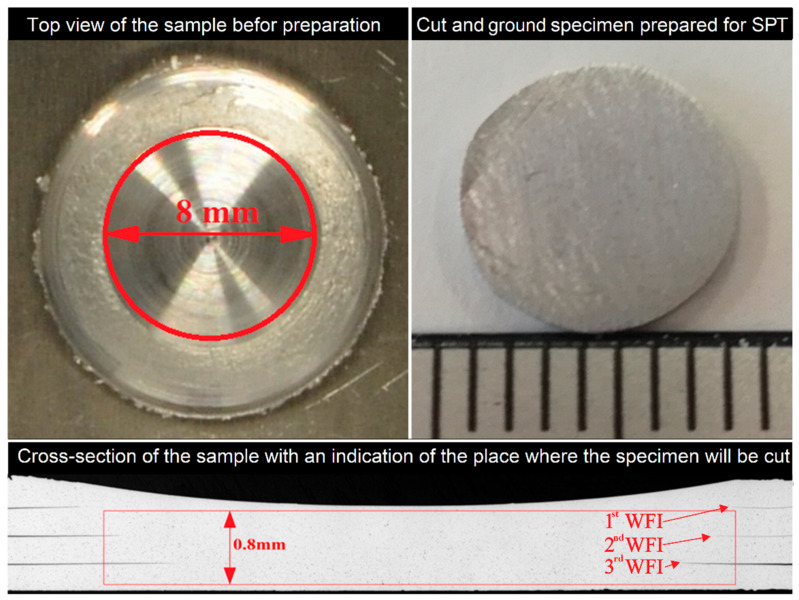
Preparation of the small punch test (SPT) specimen; the top figure shows the 8 mm diameter of the sample (width measurement in the bottom figure) such that the thickness of the sheet was 0.8 mm (WFI–weld faying interface).

**Figure 3 materials-14-01157-f003:**
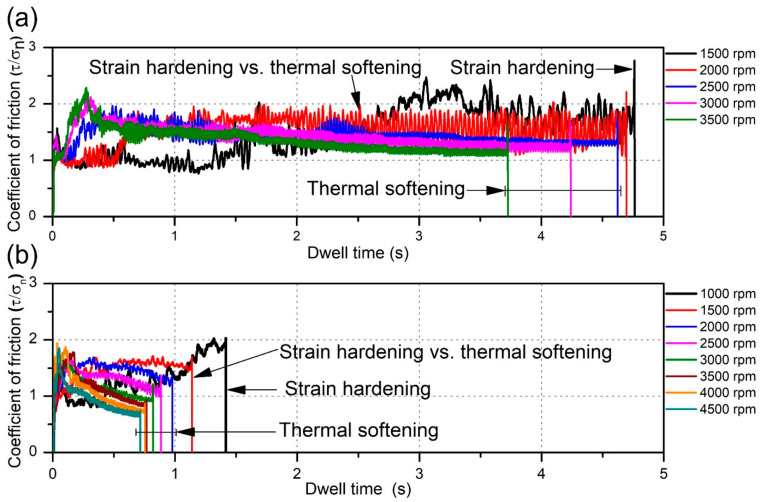
Variation in the coefficient of friction (CoF) over dwell time in: (**a**) batch I; (**b**) batch II.

**Figure 4 materials-14-01157-f004:**
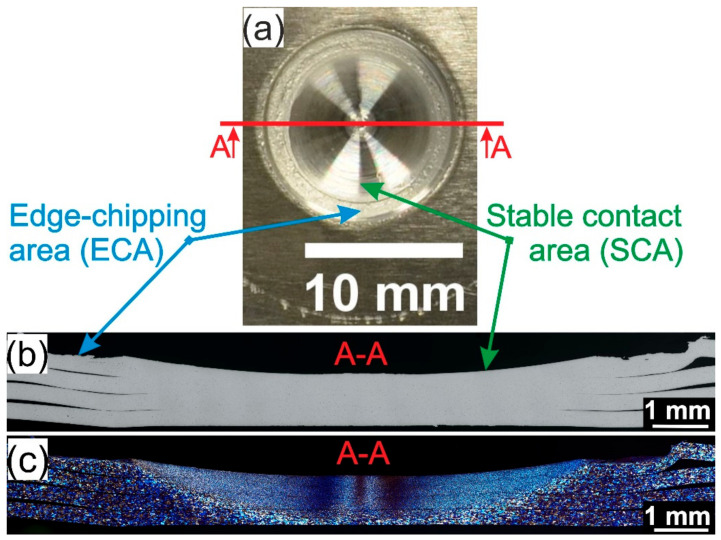
Sample welded at 1500 rpm and with tool T2 (batch II): (**a**) top view macrograph, (**b**) polished and (**c**) etched cross-section views of the welded joints.

**Figure 5 materials-14-01157-f005:**
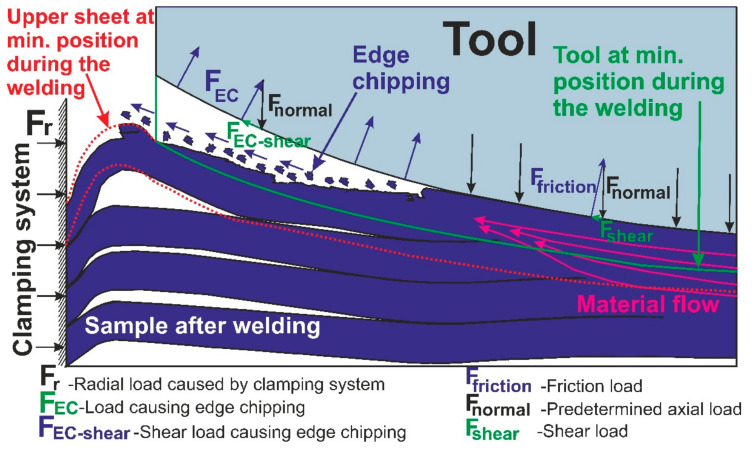
Schematic of the edge-chipping effect.

**Figure 6 materials-14-01157-f006:**
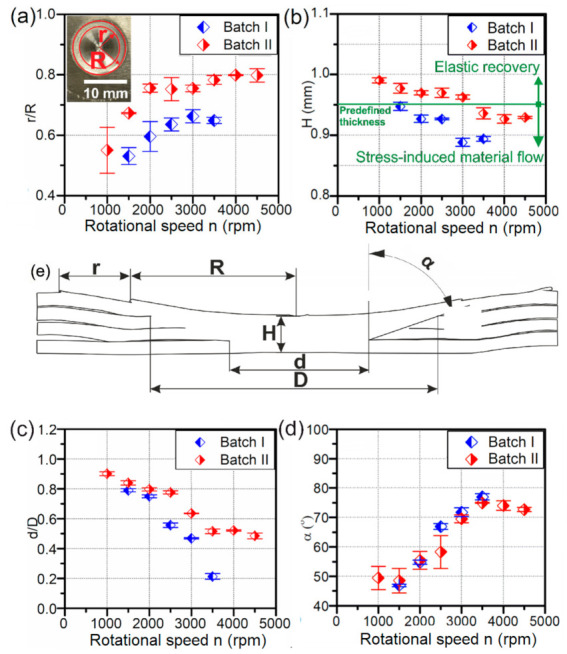
Dimensional quantification of the weld as a function of the rotational speed of the tool: (**a**) ratio of the radius of the edge-chipping area and stable process-pressure area-r/R; (**b**) axial thickness of the weld-H (**c**) ratio of the bottom diameter of the stir zone and top diameter of the stir zone-d/D; (**d**) trapper angle of the stir zone-α; (**e**) schematic of the weld.

**Figure 7 materials-14-01157-f007:**
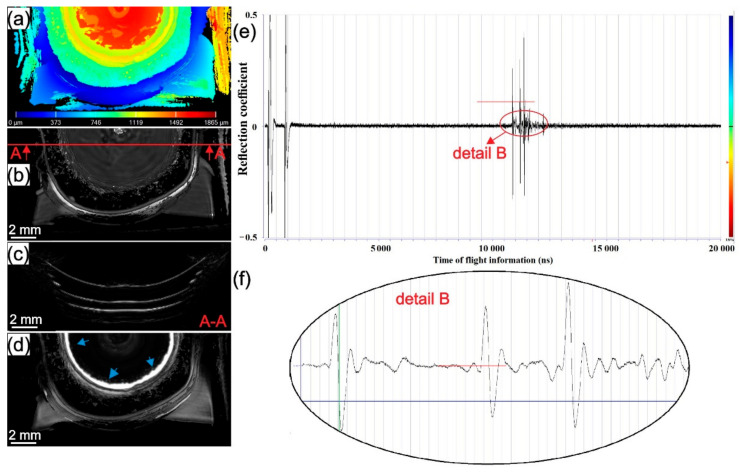
The sample from batch II welded with 1000 rpm: (**a**) topography map from the top surface; (**b**) C-scan with gate position 20 ns and gate length 50 ns with A-A cross-section showing the place where B-scan was obtained; (**c**) B-scan from A-A cross-section; (**d**) C-scan with gate position 220 ns and gate length 50 ns as shown in detail B with a red line on A-scan; (**e**) A-scan with detail showing the gate length of C-scan from (**c**); (**f**) detail of A-A scan from (**e**).

**Figure 8 materials-14-01157-f008:**
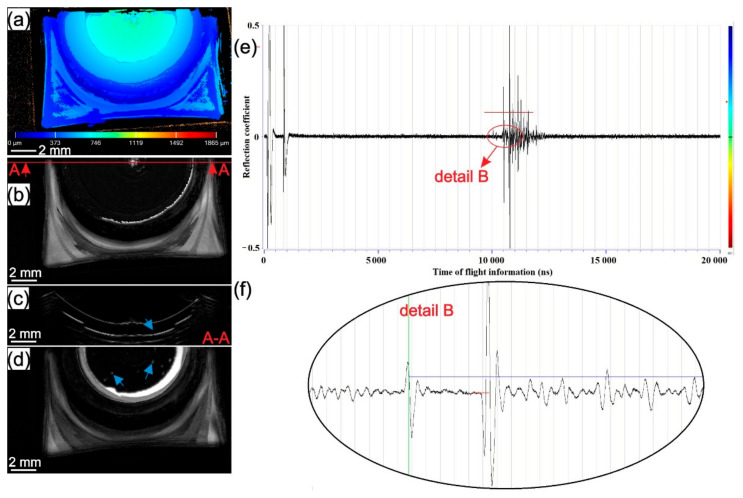
The sample from batch II welded with 2500 rpm: (**a**) topography map from the top surface; (**b**) C-scan with gate position 20 ns and gate length 50 ns with A-A cross-section showing the place where B-scan was obtained; (**c**) B-scan from A-A cross-section; (**d**) C-scan with gate position 220 ns and gate length 50 ns as shown in detail B with red line on A-scan; (**e**) A-scan with detail showing the gate length of C-scan from (**c**); (**f**) detail of A-A scan from (**e**).

**Figure 9 materials-14-01157-f009:**
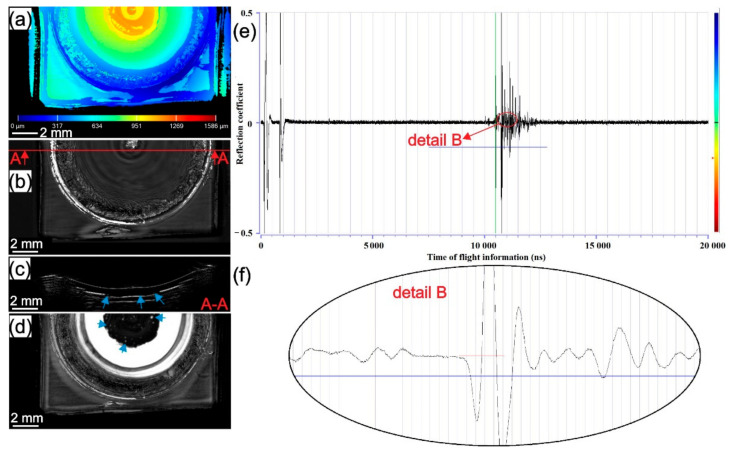
The sample from batch II welded with 4500 rpm: (**a**) topography map from the top of the sample; (**b**) C-scan with gate position 20 ns and gate length 50 ns with A-A cross-section showing the place where B-scan was obtained; (**c**) B-scan from A-A cross-section; (**d**) C-scan with gate position 220 ns and gate length 50 ns as shown in detail B with red line on A-scan; (**e**) A-scan with detail showing the gate length of C-scan from (**c**); (**f**) detail of A-A scan from (**e**).

**Figure 10 materials-14-01157-f010:**
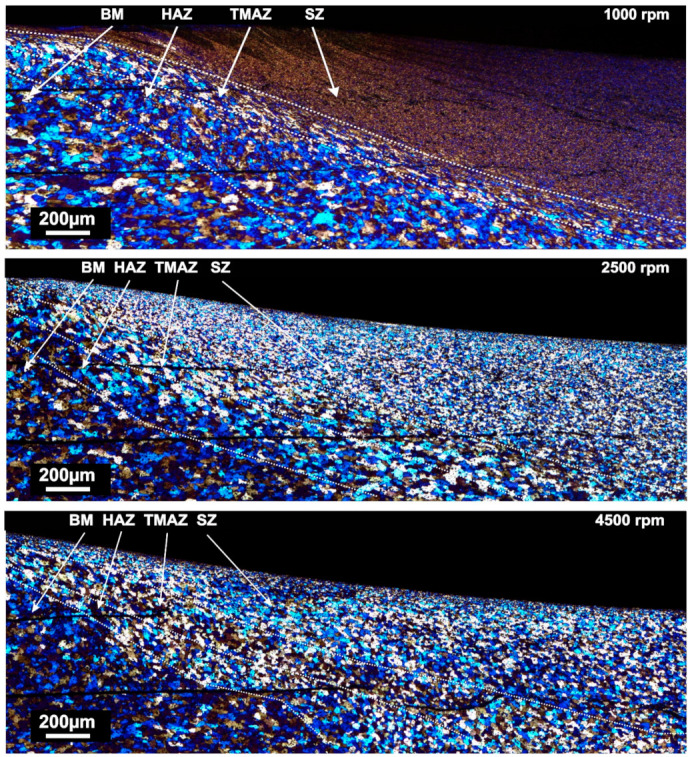
Micrographs showing the transition between the base material and stir zone for samples welded with 1000, 2500 and 4500 RPM from the batch I (BM–Base material; HAZ–Heat affected zone; TAMAZ–Thermo-mechanically affected zone; SZ–Stir zone.

**Figure 11 materials-14-01157-f011:**
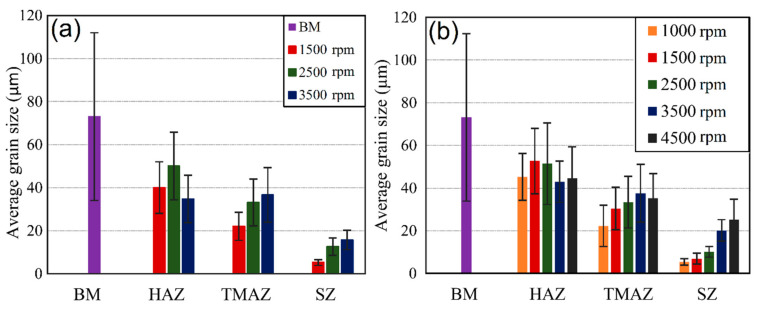
Variation in grain size between samples welded with different rotational speeds in different weld zones compared with those of the base material: (**a**) batch I [50]; (**b**) batch II (BM – Base material; HAZ–Heat affected zone; TAMAZ–Thermo-mechanically affected zone; SZ–Stir zone).

**Figure 12 materials-14-01157-f012:**
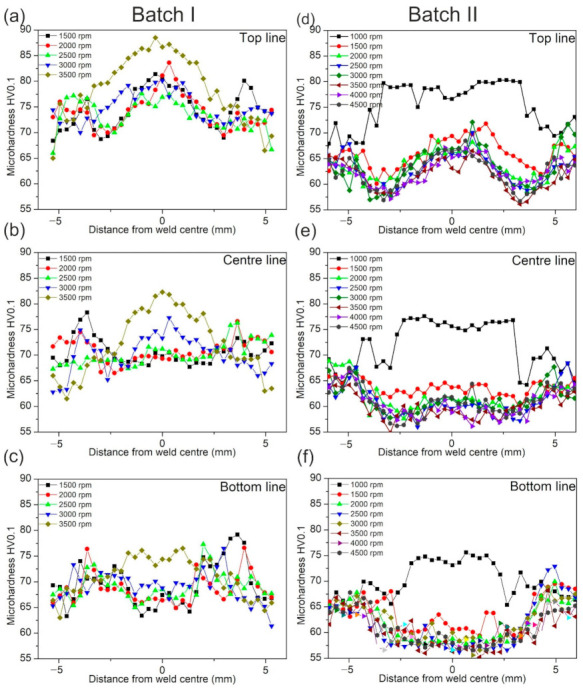
Microhardness profile measured at the weld cross-sections: (**a**) hardness profiles from the batch I in the top line (0.9 mm from the bottom of the sample); (**b**) in the middle line (0.6 mm from the bottom of the sample) (**c**) in the bottom line (0.3 mm from the bottom of the sample); (**d**) hardness profiles from batch II in top line; (**e**) in the middle line; (**f**) bottom line.

**Figure 13 materials-14-01157-f013:**
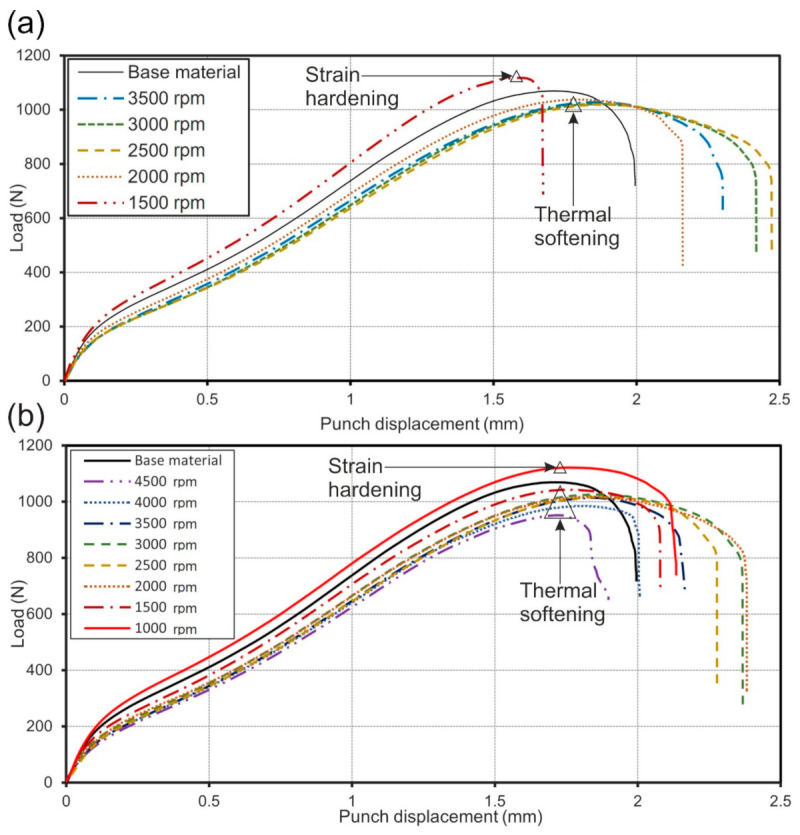
The load-displacement curves comparing the base material and weld specimens obtained with a small punch test (SPT): (**a**) batch I (1500 to 3500 rpm); (**b**) batch II (1500 to 4000 rpm).

**Figure 14 materials-14-01157-f014:**
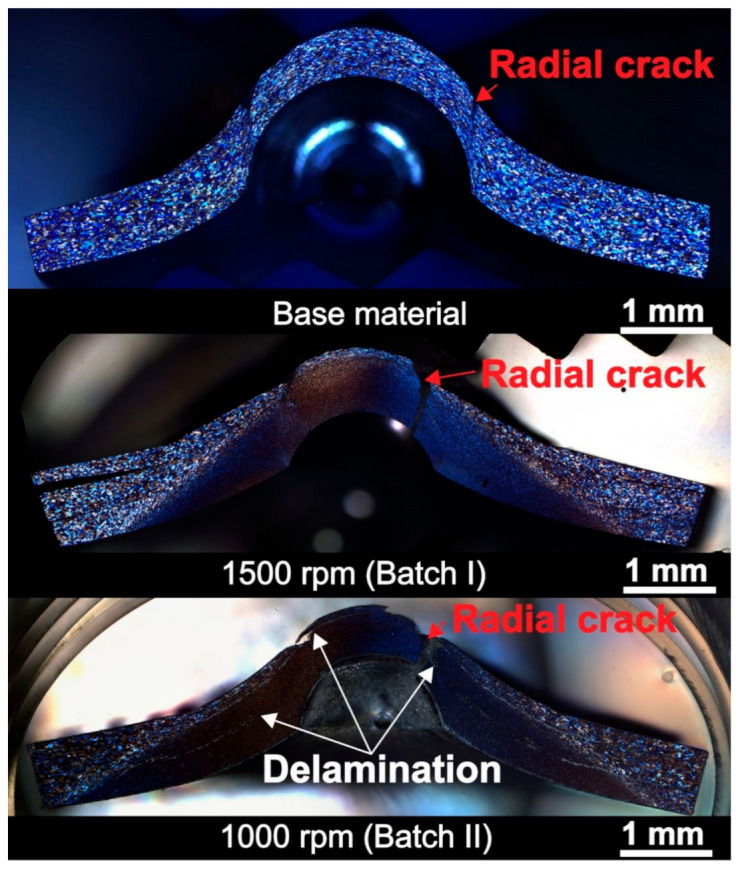
Macrographs of small punch test specimens of base material and samples welded with minimum rotational speeds from both batches (strain hardened samples).

**Figure 15 materials-14-01157-f015:**
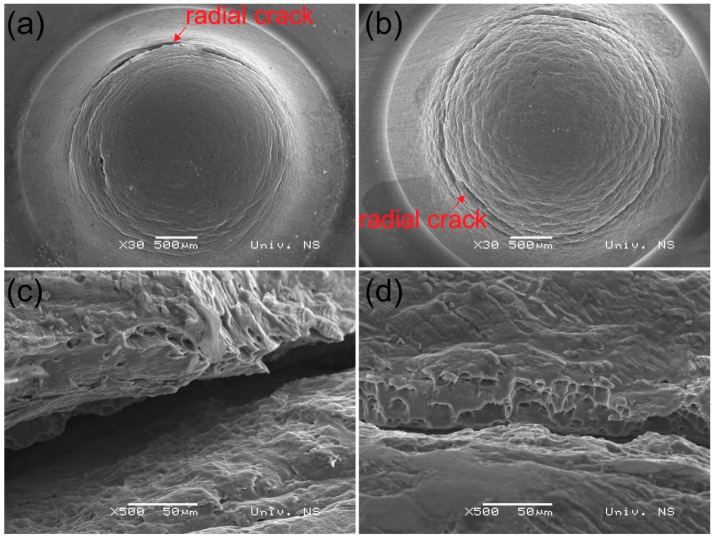
Scanning electron microscopy (SEM) images of the welds from the batch I after small punch test: (**a**) top view of the specimen obtained at 1500 rpm (RW1); (**b**) top view of the specimen obtained at 3500 rpm (RWA1); (**c**) higher magnification of the rupture–RW1; (**d**) higher magnification of the rupture–RWA1.

**Table 1 materials-14-01157-t001:** Chemical composition of AA 5754-H111 aluminium alloy used [9].

Element	Si	Fe	Cu	Mn	Mg	Cr	Zn	Ti	Al
wt. (%)	0.19	0.24	0.03	0.30	3.10	0.03	0.005	0.014	bal.

**Table 2 materials-14-01157-t002:** Mechanical properties of the AA 5754-H111 aluminium alloy used, according to the material supplier.

Material	Min. Yield Strength (MPa)	Tensile Strength (MPa)	Elongation-A50 (%)
AA 5754-H111	80	190 ÷ 240	12

**Table 3 materials-14-01157-t003:** Friction stir spot welding (FSSW) parameters.

Batch I (T1) *	Axial Load F (kN)	Rotational Speed n (rpm)	Penetration Depth h (mm)	Dwell Time td (s)	Dwell Time Deviation td (s)	Batch II (T2) *	Axial Load F (kN)	Rotational Speed n (rpm)	Penetration Depth h (mm)	Dwell Time td (s)	Dwell Time Deviation td (s)
RW1	2	1500	0.25	4.65	0.268	RWA1	4	1000	0.25	1.68	0.070
RW2	2	2000	0.25	4.60	0.340	RWA2	4	1500	0.25	1.34	0.042
RW3	2	2500	0.25	4.41	0.298	RWA3	4	2000	0.25	1.11	0.081
RW4	2	3000	0.25	4.29	0.212	RWA4	4	2500	0.25	1.07	0.076
RW5	2	3500	0.25	3.77	0.148	RWA5	4	3000	0.25	0.97	0.040
						RWA6	4	3500	0.25	1.00	0.038
						RWA7	4	4000	0.25	0.98	0.042
						RWA8	4	4500	0.25	0.91	0.015

**Table 4 materials-14-01157-t004:** Rupture loads (Fm) with standard deviations used in calculations and weld efficiency (Es) of the welds from the batch I and II.

	Base Material	4500 rpm	4000 rpm	3500 rpm	3000 rpm	2500 rpm	2000 rpm	1500 rpm	1000 rpm
**Batch I**									
Rupture Load Fm (N)	1094.9			1004	1030.2	1010.3	1043	1162	
Standard Deviation	21.3			37.83	3.69	4.14	58.01	9.51	
Joint Efficiency E (%)	−			91.7	94.1	92.3	95.3	106.1	
**Batch II**									
Rupture Load Fm (N)	1094.9	957.2	991.9	1002.9	1001.9	1021.3	1016.6	1040.3	1096.7
Standard Deviation	21.3	10.3	5.7	13.4	17.4	23.3	3.6	1.9	33.5
Joint efficiency E (%)	−	87.4	90.6	91.6	91.5	93.3	92. 8	95	100.2

**Table 5 materials-14-01157-t005:** The specimen thinning (measured from the macrographs as shown in Figure 14 in the middle of the stir zone for each specimen from the batch I and II and then subtracted from an initial value of specimen thickness 0.8 mm).

Specimen Thinning		4500 rpm	4000 rpm	3500 rpm	3000 rpm	2500 rpm	2000 rpm	1500 rpm	1000 rpm
Batch I	(mm)			0.35 ± 0.01	0.48 ± 0.03	0.39 ± 0.02	0.32 ± 0.01	0.12 ± 0.01	
	(%)			43.8 ± 1.25	60 ± 3.75	48.8 ± 2.5	40 ± 2.5	15 ± 1.25	
Batch II	(mm)	0.19 ± 0.03	0.36 ± 0.02	0.28 ± 0.03	0.29 ± 0.01	0.26 ± 0.02	0.26 ± 0.02	0.28 ± 0.01	0.17 ± 0.01
	(%)	23.8 ± 3.75	45 ± 2.5	35 ± 3.75	36.3 ± 1.25	32.5 ± 2.5	32.5 ± 2.5	35 ± 1.25	21.3 ± 1.25
Base material	(mm)	0.27 ± 0.3							
	(%)	33.75 ± 3.75							

## Data Availability

The data presented in this study are available on request from the corresponding author.

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
