# Peer review of "Influence of Tool Geometry and Process Parameters on the Properties of Friction Stir Spot Welded Multiple (AA 5754 H111) Aluminium Sheets"

_materials, 2021, doi:10.3390/ma14051157_

Round 1
Reviewer 1 Report
The reviewer comments of the paper «Influence of tool geometry and process parameters on properties of friction stir spot welded multiple AA 5754 aluminium sheets»
- Reviewer
The authors presented an article «Influence of tool geometry and process parameters on properties of friction stir spot welded multiple AA 5754 aluminium sheets». However, there are several points in the article that require further explanation.
Comment 1:
Overall, the introduction is well written.
However, explain why the material chosen for research is so important for the study.
It is useful to add an article: Effect of Feed Rate in FSW on the Mechanical and Microstructural Properties of AA5754 Joints. doi:10.1155/2019/4156176
Comment 2:
- Materials and Methods
Are the formulas given in the article original? If not needed appropriate citations.
Are all figures original? If not needed appropriate citations and permissions.
Figure 4 title is split. Eliminate this shortcoming.
Figure 11. Sign the title of the horizontal axis. And in the figure caption, decipher all the abbreviations. This should be clear to the reader.
Make the names of the axes of the figures "complete", for example, instead of "n (rpm)" use "Rotational speed n (rpm)". Apply this to all figures.
Apply the same to tables.
Comment 3:
It will be useful to add a section of Nomenclature in which to sign all the physical quantities and abbreviations encountered in the article. There are many physical quantities in the text and such a section will help to find the description of the necessary element.
For example,
n : Rotational speed (rpm)
FSSW : Friction stir spot welding
etc.
The article is interesting and written at a good scientific level. Authors should carefully study the comments and make improvements to the article. After major changes can an article be considered for publication in the «Materials».
Reviewer 2 Report
In the work, four ultrathin sheets 0.3 mm thick of AA5754-H111 alloy were welded by FSSW. The influence of tool geometries and process parameters was studied. It will interest researchers in the field of lap-welding of multiple ultrathin sheets. Before being considered favorably for publication, there are several details should be addressed.
- For joining the ultrathin sheets 0.3mm thickness, why chose so big tool radius? Why the tools are convex? How about concave just like FSW tool? What is the role? How about the smaller one?
- In the part of Abstract and Conclusion, Sound and concise conclusions with the supporting results should be given to the reader.
- As the authors claimed that the work focuses on the automotive and aerospace industry, why chose lap-welding of multiple ultrathin sheets? For battery components, stand-thermal connectors, terminals and wire conductors, is four ultrathin sheets enough? Is 0.3mm thickness thin enough? 0.1mm maybe good.
- FSSW is evolved as a solid-state processing technique derived from friction stir welding. The recent progress on control strategies for inherent issues in friction stir welding/processing were suggested to add, such as https://doi.org/10.1016/j.pmatsci.2020.100752 and https://doi.org/10.1016/j.pmatsci.2020.100706.
Reviewer 3 Report
Dear Authors,
I have reviewed paper "Influence of tool geometry and process parameters on properties of friction stir spot welded multiple AA 5754 aluminium sheets". It fulfills the aims and scope of the journal, and could be considered for publishing after some minor improvements.
My overal merit about this work is very high. Congratulations.
Introduction:
- This part is really well written. Congratulations.
- At the and of introduction please clearly mark the novelty of your work.
Materials and Methods:
- Please add information about mechanical properties (Re, Rm and A) of used material.
- Have you used any standards and their requirements during tests? If yes, please mention these standards. If not, please describe why.
Results:
- In my opinion, the name of this paragraph should be changed to "Results and Discussion". You have discussed your results together. Also, you compared them with relevant literature, so the phrase "Discussion" should be added.
- Please add reference to equation (2).
- Fig. 4 - the name of this figure was cut "pol-". Something moved in your manuscript. Between lines 240 and 241 the next fig. appeared. Please improve.
- Fig. 7, 8 and 9 - please add scale bars.
- Line 469 - this is not equation (2), this is (3). Equation (2) was presented in line 196.
- Please add units to the each ;art of presented equations.
Conclusions:
- This part is well written. Conclusions are strongly connected with results.
Round 2
Reviewer 1 Report
The authors have improved the article according to the comments. The article can now be published.
Reviewer 2 Report
The authors have made point-by-point responses to the comments. The modifications are satisfied. It is acceptable for publication.